# A Quadruplex qRT-PCR for Differential Detection of Four Porcine Enteric Coronaviruses

**DOI:** 10.3390/vetsci9110634

**Published:** 2022-11-16

**Authors:** Hongjin Zhou, Kaichuang Shi, Feng Long, Kang Zhao, Shuping Feng, Yanwen Yin, Chenyong Xiong, Sujie Qu, Wenjun Lu, Zongqiang Li

**Affiliations:** 1College of Animal Science and Technology, Guangxi University, Nanning 530005, China; 2Guangxi Center for Animal Disease Control and Prevention, Nanning 530001, China

**Keywords:** porcine enteric coronavirus, porcine epidemic diarrhea virus (PEDV), transmissible gastroenteritis virus (TGEV), porcine deltacoronavirus (PDCoV), swine acute diarrhea syndrome coronavirus (SADS-CoV), multiplex qRT-PCR, detection method

## Abstract

**Simple Summary:**

Porcine epidemic diarrhea virus (PEDV), transmissible gastroenteritis virus (TGEV), porcine deltacoronavirus (PDCoV), and swine acute diarrhea syndrome coronavirus (SADS-CoV) cause similar manifestations of diarrhea, vomiting, and dehydration. In this study, a quadruplex real-time quantitative PCR (qRT-PCR) assay was developed for the differential detection of PEDV, TGEV, PDCoV, and SADS-CoV. A total of 3236 clinical fecal samples from Guangxi province, China, were tested to evaluate the application of the quadruplex qRT-PCR, and the positive rates of PEDV, TGEV, PDCoV, and SADS-CoV were 18.26%, 0.46%, 13.16%, and 0.15%, respectively. The developed assay showed extreme specificity, high sensitivity, and excellent reproducibility for the simultaneous detection and differentiation of PEDV, TGEV, PDCoV, and SADS-CoV.

**Abstract:**

Porcine epidemic diarrhea virus (PEDV), transmissible gastroenteritis virus (TGEV), porcine deltacoronavirus (PDCoV), and swine acute diarrhea syndrome coronavirus (SADS-CoV) are four identified porcine enteric coronaviruses. Pigs infected with these viruses show similar manifestations of diarrhea, vomiting, and dehydration. Here, a quadruplex real-time quantitative PCR (qRT-PCR) assay was established for the differential detection of PEDV, TGEV, PDCoV, and SADS-CoV from swine fecal samples. The assay showed extreme specificity, high sensitivity, and excellent reproducibility, with the limit of detection (LOD) of 121 copies/μL (final reaction concentration of 12.1 copies/μL) for each virus. The 3236 clinical fecal samples from Guangxi province in China collected between October 2020 and October 2022 were evaluated by the quadruplex qRT-PCR, and the positive rates of PEDV, TGEV, PDCoV, and SADS-CoV were 18.26% (591/3236), 0.46% (15/3236), 13.16% (426/3236), and 0.15% (5/3236), respectively. The samples were also evaluated by the multiplex qRT-PCR reported previously by other scientists, and the compliance rate between the two methods was more than 99%. This illustrated that the developed quadruplex qRT-PCR assay can provide an accurate method for the differential detection of four porcine enteric coronaviruses.

## 1. Introduction

Coronaviruses (CoVs) in the *Coronaviridae* family are enveloped RNA viruses with single-stranded, positive-sense genomes [1], and they cause gastrointestinal and respiratory diseases in animals and humans [2]. To date, six CoVs have been identified as causing respiratory illness or gastroenteritis in pigs, which include transmissible gastroenteritis virus (TGEV), porcine hemagglutinating encephalomyelitis virus (PHEV), porcine epidemic diarrhea virus (PEDV), porcine respiratory coronavirus (PRCoV), porcine deltacoronavirus (PDCoV), and swine acute diarrhea syndrome coronavirus (SADS-CoV) [3]. Of these, the four porcine enteric coronaviruses, PEDV, TGEV, PDCoV, and SADS-CoV, have been reported to cause acute gastroenteritis in neonatal piglets, characterized by loss of weight, diarrhea, vomiting, and dehydration [4,5]. In 1971, PEDV was first identified in England, and has been reported worldwide since then [6]. In 2010, PED, caused by a novel mutant PEDV strain, outbroke in China, which resulted in an almost 100% morbidity rate and about an 80–100% mortality rate in neonatal piglets under seven days of age [7]. In 2013, the mutant PEDV strain was also discovered in the United States, and, subsequently, in other countries of America, Asia, and Europe [8,9,10]. TGEV was first reported in the United States in 1946, and has been identified in America, Europe, Asia, and Africa [4,11]. TGEV was first identified in China in the 1980s, and has still been circulating in recent years [12,13]. In 2017, SADS-CoV was first found in suckling piglets with acute diarrhea symptoms in Guangdong province, Southern China [14]. Subsequently, SADS-CoV was found in Guangdong, Fujian, and Guangxi provinces in Southern China, while no SADS-CoV has been discovered in other countries until now [15,16,17]. A new porcine deltacoronavirus (PDCoV) called HKU15 coronavirus was found in Hong Kong, China, in 2009 [18]. In 2014, PDCoV was reported to be associated with an outbreak of severe acute pig diarrhea in the United States [19], and was subsequently identified in diarrheal piglets in major pig breeding countries around the world [20,21,22]. These enteric CoVs have caused huge damage to the swine industry throughout the world [11,23].

PEDV, TGEV, PDCoV, and SADS-CoV cause similar signs of diarrhea, vomiting, and dehydration in neonatal piglets and it is hard to differentiate by clinical manifestations [4,5]. Furthermore, co-infection of two or more of these viruses has been reported in some pig farms [13,24,25]. Thus, it is necessary to differentiate these diseases through laboratory diagnosis since it is hard to differentiate these diseases based solely on clinical manifestations. To date, the multiplex RT-PCR [26,27,28,29,30,31,32,33] and the multiplex quantitative RT-PCR (qRT-PCR) [34,35,36,37,38,39] are two common methods that have been used in the laboratory for the differential detection of two or more than two of the four porcine enteric CoVs, and also used for differentiation of the porcine enteric CoVs and other circulating porcine viruses. Of the previous reports, there were two reports on the multiplex qRT-PCR for the differentiation of PEDV, TGEV, PDCoV, and SADS-CoV [37,39], and one report used PEDV M, TGEV N, PDCoV M, and SADS-CoV N genes as target templates, while the other report used PEDV N, TGEV S, PDCoV N, and SADS-CoV N genes as target templates. The qRT-PCR has advantages of a low chance of contamination, fast reaction speed, and high sensitivity, and has been widely used for the detection of viral nucleic acids. In this study, a quadruplex qRT-PCR based on PEDV N, TGEV M, PDCoV M, and SADS-CoV N genes was developed for the differential detection of the four porcine enteric CoVs, and its application was evaluated for clinical samples.

## 2. Materials and Methods

### 2.1. Vaccine Strains and Clinical Samples

The vaccine strains were as follows: CV777 strain of PEDV, H strain of TGEV, NX strain of porcine rotavirus (PoRV), O strain of foot-and-mouth disease virus (FMDV), C strain of classical swine fever virus (CSFV), TJM-F92 strain of porcine reproductive and respiratory syndrome virus (PRRSV), Bartha-K61 strain of pseudorabies virus (PRV), and SX07 strain of porcine circovirus type 2 (PCV2), which came from commercial attenuated live or inactivated vaccines. The positive samples of African swine fever virus (ASFV), PCV1, and PDCoV came from the clinical samples and were confirmed by genetic sequencing. The vaccine solutions and the clinical positive samples were used to construct the standard recombinant plasmids of PEDV, TGEV, and PDCoV or used as viral control samples in the specificity test during experiments.

All 3236 clinical fecal samples from 218 different pig farms in Guangxi province, Southern China, were collected from diarrheal piglets between October 2020 and October 2022. All clinical samples and vaccine strains were stored at −80 °C.

### 2.2. Design of Primers and Probes

Based on the genomic sequences of PEDV (AF353511), TGEV (DQ811789), PDCoV (JQ065042), and SADS-CoV (FJ617209) published in NCBI GenBank, the primers and TaqMan probes were designed to use the conserved regions of the N gene of PEDV and SADS-CoV, and the M gene of TGEV and PDCoV as the target templates, respectively (Table 1).

### 2.3. Nucleic Acid Extraction

The clinical fecal samples were resuspended with phosphate-buffered saline (PBS, pH 7.2) (in a ratio of 1:4, *W*/*V*), vortexed (5 min), and centrifuged at 4 °C (12,000 rpm, 10 min) to obtain the supernatants for extraction of total DNA and RNA, and then stored at –80 °C until used to test PEDV, TGEV, PDCoV, and SADS-CoV.

### 2.4. Construction of the Standard Plasmids

The PEDV and TGEV vaccines, PDCoV positive samples, synthesized plasmid containing SADS-CoV N gene (TaKaRa, Dalian, China) were used to construct the standard plasmids according to the procedure reported by Liu et al. [40] with some modifications. The standard plasmids were named p-PEDV, p-TGEV, p-PDCoV, and p-SADS-CoV, respectively. The following formula was used to determine their concentration: plasmid (copies/μL) = $\frac{\left(6.02\times 10^{23}\right)\times \left(X\;ng/\mu L\times 10^{-9}\right)}{plasmid\;length\;\left(bp\right)\times 660}$.

### 2.5. Optimization of the Multiplex qRT-PCR

All qRT-PCR experiments were performed using the Q5 qPCR system (ABI, Carlsbad, CA, USA). The optimal conditions were determined by altering the annealing temperature (56 °C–62 °C) and the concentration of each primer and probe (0.2–0.5 μL) (20 pmol/μL). The reaction systems of the multiplex qRT-PCR contained: 10 μL 2 × TaKaRa One-Step RT-PCR Buffer III, 0.4 μL TaKaRa Ex Taq HS (5 U/μL), 0.4 μL TaKaRa PrimeScript RT Enzyme Mix II (RNA/DNA), 0.2–0.3 μL primer mixtures [PEDV(N)-U/D, TGEV(M)-U/D, PDCoV(M)-U/D, and SADS-CoV(N)-U/D] (20 pmol/μL), 0.2–0.3 μL probe mixtures [PEDV(N)-P, TGEV(M)-P, PDCoV(M)-P, and SADS-CoV(N)-P] (20 pmol/μL), 2 μL templates, and nuclease-free distilled water to a final volume of 20 μL. The following parameters were used: 42 °C 5 min, 95 °C 10 s, and then 40 cycles of 95 °C 5 s, 57 °C 34 s. At the end of each cycle, the fluorescence signals were automatically recorded. The optimal conditions were determined to obtain the maximum ΔRn and the minimal cycle threshold (Ct) value.

### 2.6. Construction of Standard Curves

The four standard plasmids were adjusted to 1.21 × 10^9^ copies/μL, mixed together at a ratio of 1:1:1:1, and ten-fold serially diluted (1.21 × 10^9^–1.21 × 10^2^ copies/μL). Then, 2 μL of each concentration of the mixed plasmids were used as a template and amplified according to the optimized quadruplex qRT-PCR reaction conditions to construct a standard curve.

### 2.7. Specificity Analysis

The RNA or DNA of PEDV, TGEV, PDCoV, SADS-CoV, ASFV, FMDV, CSFV, PRRSV, PRV, PoRV, PCV1, and PCV2 were used as templates, and the four standard plasmids and distilled water were used as positive and negative controls for amplification to evaluate the specificity of the quadruplex qRT-PCR.

### 2.8. Sensitivity Analysis

The four standard plasmids were mixed together with a ratio of 1:1:1:1, then ten-fold serially diluted. The plasmid mixtures from 1.21 × 10^9^ to 1.21 × 10^0^ copies/μL (final reaction concentrations ranging from 1.21 × 10^8^ to 1.21 × 10^−1^ copies/μL) were used as templates for amplification to determine the limit of detection (LOD) of each plasmid for the quadruplex qRT-PCR.

### 2.9. Repeatability Analysis

The repeatability of the quadruplex qRT-PCR was determined by evaluating the coefficients of variance (CVs) of the intra-assay and inter-assay. The four standard plasmids were mixed together with a ratio of 1:1:1:1 and ten-fold serially diluted, and the plasmids of 1.21 × 10^9^, 1.21 × 10^7^, 1.21 × 10^5^ copies/μL (1.21 × 10^8^, 1.21 × 10^5^, 1.21 × 10^4^ copies/μL for final reaction concentrations) were used as templates. For the intra-assay of repeatability, all templates were performed in triplicate. For the inter-assay of reproducibility, all templates were performed on three different days.

### 2.10. Detection of the Clinical Samples

The 3236 clinical fecal samples were resuspended, vortexed, and centrifuged. Then, the total nucleic acids were extracted, and used to test PEDV, TGEV, PDCoV, and SADS-CoV by the developed quadruplex qRT-PCR. At the same time, these samples were also tested by the multiplex qRT-PCR developed by Huang et al. with some modifications [37], and the compliance rate of these two methods was further evaluated.

## 3. Results

### 3.1. Construction of Standard Plasmids

After PCR amplification, the purified target fragments of PEDV N, TGEV M, PDCoV M, and SADS-CoV N genes were used to construct the standard plasmids. Finally, the original concentrations of the standard plasmids named p-PEDV, p-TGEV, p-PDCoV, and p-SADS-CoV were 3.53 × 10^10^, 1.34 × 10^10^, 2.13 × 10^10^, and 1.21 × 10^10^ copies/μL, respectively.

### 3.2. Determination of the Optimal Reaction Conditions

After optimizing the annealing temperature, and the concentrations of primer and probe, the total 20 μL volume of the optimal reaction system for the quadruplex qRT-PCR was determined (Table 2). The following amplified parameters were obtained: 42 °C for 5 min, 95 °C for 10 s, and then 40 cycles of 95 °C for 5 s and 57 °C for 34 s. The fluorescence signals were recorded for each cycle. The cutoff Ct value of 36 cycles was set for a positive sample.

### 3.3. Generation of Standard Curves

The four standard plasmids were mixed together with a ratio of 1:1:1:1, ten-fold serially diluted, and amplified with the optimized quadruplex qRT-PCR system at a concentration of 1.21 × 10^9^ to 1.21 × 10^2^ copies/μL (final reaction concentration: 1.21 × 10^8^ to 1.21 × 10^1^ copies/μL). The standard curves were generated, indicating that the slope of equation, correlation coefficient (R^2^), and amplification efficiency (E) were −3.130, 0.999, and 108.684% for PEDV; −3.148, 0.999, and 107.798% for TGEV; −3.189, 0.999, and 105.858% for PDCoV; and −3.100, 0.999, and 110.185% for SADS-CoV, respectively (Figure 1).

### 3.4. Specificity Analysis

Different porcine viruses were tested to evaluate the specificity of the quadruplex qRT-PCR. The results illustrated that only PEDV, TGEV, PDCoV, and SADS-CoV recorded corresponding signals and generated specific amplification curves, while ASFV, FMDV, CSFV, PRRSV, PRV, PoRV, PCV1, and PCV2, could not detect any positive signals and did not demonstrate any amplification curves (Figure 2), indicating the extreme specificity of the quadruplex qRT-PCR.

### 3.5. Sensitivity Analysis

The four standard plasmids were mixed together with a ratio of 1:1:1:1, ten-fold serially diluted from 1.21 × 10^9^ to 1.21 × 10^0^ copies/μL (final reaction concentrations: 1.21 × 10^8^ to 1.21 × 10^−1^ copies/μL), and the sensitivity of the assay was determined. A Ct value higher than 36 cycles was considered as a negative sample. The LOD of each of PEDV, TGEV, PDCoV, and SADS-CoV was 1.21 × 10^2^ copies/μL (final reaction concentration: 1.21 × 10^1^ copies/μL) (Figure 3), indicating the high sensitivity of the quadruplex qRT-PCR.

### 3.6. Repeatability Analysis

To estimate the repeatability of the assay, three plasmid mixtures of 1.21 × 10^9^, 1.21 × 10^7^, and 1.21 × 10^5^ copies/μL (final reaction concentrations: 1.21 × 10^8^, 1.21 × 10^6^, and 1.21 × 10^4^ copies/μL) were used to determine the intra-assay and inter-assay variation. The coefficients of variation (CVs) of the intra-assay, and the inter-assay for all the four plasmids ranged from 0.16% to 1.48%, and from 0.34% to 1.87%, respectively (Table 3), indicating the excellent repeatability of the quadruplex qRT-PCR.

### 3.7. Detection Results of the Clinical Samples

The developed assay was used to evaluate the 3236 clinical fecal samples from Guangxi province, China. The positive rates of PEDV, TGEV, PDCoV, and SADA-CoV were 18.26% (591/3236), 0.46% (15/3236), 13.16% (426/3236), and 0.15% (5/3236), respectively (Table 4). Furthermore, the co-infection rates of PEDV/TGEV and PEDV/PDCoV were 0.06% (2/3236) and 1.42% (46/3236), respectively (Table 4).

All of the 3236 clinical samples were also evaluated by the multiplex qRT-PCR established by Huang et al. [37], and the positive rates of PEDV, TGEV, PDCoV, and SADS-CoV were 17.67% (572/3236), 0.46% (15/3236), 12.89% (417/3236), and 0.15% (5/3236), respectively. The coincidence rates between these two methods were 99.41%, 100%, 99.72%, and 100%, respectively (Table 5).

## 4. Discussion

PEDV, TGEV, PDCoV, and SADS-CoV cause similar symptoms of dehydration, diarrhea, vomiting, and loss of weight in infected piglets [4,5], and the accurate differential diagnosis of these diseases has depended on laboratory testing and diagnosis. The multiplex RT-PCR/qRT-PCR have been reported for the differential detection of two or more than two of these four porcine enteric CoVs [26,27,28,29,30,31,32,33,34,35,36,37,38,39]. In our developed quadruplex qRT-PCR, the specific primers and probes targeted the conserved regions of PEDV N, TGEV M, PDCoV M, and SADS-CoV N genes, respectively. The assay could specifically detect PEDV, TGEV, PDCoV, and SADS-CoV, and had no cross-reaction with other porcine viruses that are circulating in Chinese pig herds at present. The LOD was 121 copies/μL (final reaction concentration of 12.1 copies/μL) for four standard plasmids, and the intra- and inter-assay CVs were between 0.16% and 1.87%. These showed the extreme specificity, high sensitivity, and excellent repeatability of the assay. Furthermore, to validate its application in the field, the 3236 clinical samples were tested by the developed assay. Meanwhile, these 3236 clinical samples were also tested by the multiplex qRT-PCR established by Huang et al. [37], and the results indicated that the two methods had a compliance rate of more than 99%, which confirmed the usefulness of the developed quadruplex qRT-PCR. This assay used the combination of PEDV N, TGEV M, PDCoV M, and SADS-CoV N genes, which was different from the combination of the different genes in the previous reports [37,39], and provided a new choice for the detection of these four porcine enteric coronaviruses.

The 3236 field samples from Guangxi province between October 2020 and October 2022 were tested by the developed assay. The positive rates of PEDV, TGEV, PDCoV, and SADS-CoV were 18.26% (591/3236), 0.46% (15/3236), 13.16% (426/3236), and 0.15%, respectively, indicating that PEDV and PDCoV were the predominant porcine enteric CoVs circulating in Guangxi province, and TGEV still occasionally occurred in pig herds, while positive samples of SADS-CoV were also found in three pig farms in 2022. It was noteworthy that the co-infection rates of PEDV/PDCoV and PEDV/TGEV were 1.42% (46/3236), and 0.06% (2/3236), respectively, indicating that PEDV and PDCoV co-infection remained prevalent in pig herds in Guangxi province. Recently, some reports showed that PEDV, TGEV, and PDCoV were common in many pig herds in China, and PEDV was usually more frequently found than other porcine enteric CoVs, while the co-infection of PEDV/PDCoV was the most common [13,24,32,33,36,37,38,39,41]. For example, Li et al. reported that 7107 samples from South–Central China showed positive rates of 56.13%, 0.91%, 8.24%, and 2.11% for PEDV, TGEV, PoRV-A, and PDCoV, respectively, and co-infection rates of 1.36%, 0.31%, and 0.04% for PEDV and PoRV-A, PEDV and PDCoV, and PEDV and TGEV, respectively [13]. Zhang et al. reported that 2987 samples from Eastern–Southern China showed positive rates of 57.32%, 27.22%, 0.70%, 0.84%, and 0.23% for PEDV, PDCoV, TGEV, PoRV, and SADS-CoV, respectively, and PEDV/PDCoV co-infection was the most frequent (12.72%) [24]. Si et al. reported that 181 samples from Southern China showed positive rates of 30.94%, 17.67%, 11.6%, 9.39%, and 0.55% for PEDV, PDCoV, SADS-CoV, PoRV-A, and TGEV, respectively, with the highest co-infection rate (9.39%) of PEDV/PDCoV [32]. Compared with PEDV and PDCoV, the positive rate of TGEV was relatively low in many provinces in China [13,24,32,39], but more attention still needs to be paid to this virus since it might cause high morbidity and mortality [42]. Furthermore, since co-infections of different enteric CoVs enhance the disease severity in piglets, aggravating the diseases and resulting in heavy losses [43,44], great attention should be attached to the harm of the co-infection of porcine enteric CoVs to the pig industry.

From October 2016 to May 2017, SADS was observed in Guangdong province, Southern China, with up to 90% mortality for piglets younger than 5 days old. The causative agent, SADS-CoV, was identified in 2017 [14]. SADS-CoV was subsequently reported in Fujian province in 2018 [24], and in Guangxi province in 2021 [17]. To date, SADS-CoV has only been reported in Guangdong, Fujian, Liaoning, Gansu, and Guangxi provinces of China, and not yet discovered in other countries [16,17,24,37,45]. However, the disease re-emerged in Guangdong province one year after the first outbreak [46]. Therefore, this virus needs to be continuously surveyed in Guangxi province in case of it re-emerging. In our study, the 3236 clinical samples from Guangxi province were evaluated for SADS-CoV, and 0.15% (5/3236) of the samples were found to be positive for SADS-CoV, indicating that SADS-CoV has been circulating in some pig herds since it was first identified in Guangxi province in 2021, even if the virus has not yet caused massive piglet diarrhea in Guangxi province since the outbreak of SADS in Guangdong province in 2017. Furthermore, since Guangxi province is located to the west of Guangdong province and has a border of nearly 1000 km with Guangdong province, and a huge amount of people and vehicles travel between the two provinces, and with a large volume of animal and animal product trades, there is a great risk of SADS-CoV spreading from Guangdong to Guangxi province, which raises more than 40-million pigs each year. Therefore, it is necessary to strengthen monitoring, ensure a timely grasp of any epidemic situation, and provide basic data for the accurate prevention and control of SADS, and this developed assay can provide a specific, sensitive, and accurate method for testing this pathogen.

It is noteworthy that only ASFV, FMDV, CSFV, PRRSV, PRV, PoRV, PCV1, and PCV2 were used to evaluate the specificity of the developed quadruplex qRT-PCR in this study. These viruses are the common pathogens circulating in the field in China nowadays, so they were used as the control viruses for specificity analysis of the developed assay. Other commonly circulating viruses, for example, hepatitis E virus (HEV), porcine kobuvirus, porcine astrovirus, and porcine torovirus, were not used to evaluate the specificity of the developed assay. The other issue is that genetic diversity of these viruses exists in the field, and these viruses should be consistently monitored to obtain information on their genetic variation promptly in order to adjust the primers and probes timely according to the genetic variation in the circulating viruses, which will ensure that the established method can accurately detect the clinical epidemic strains.

## 5. Conclusions

PEDV, TGEV, PDCoV, and SADS-CoV are the important porcine enteric coronaviruses that seriously threaten the pig industry all over the world. In this study, a quadruplex qRT-PCR was developed to differentially detect these four viruses, with extreme specificity, high sensitivity, and excellent repeatability. The assay could be used to test the four pathogens in one reaction at the same time and was an efficient method to simultaneously detect and differentiate PEDV, TGEV, PDCoV, and SADS-CoV.

## Figures and Tables

**Figure 1 vetsci-09-00634-f001:**
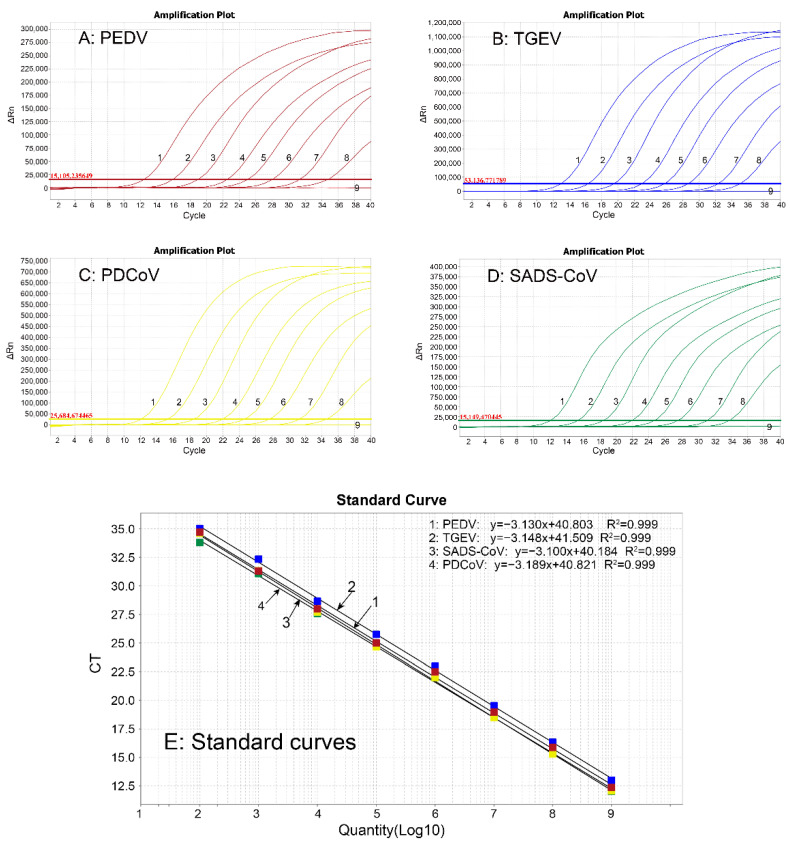
Generation of the standard curves. The amplification curves were generated from the standard plasmid p-PEDV (**A**), p-TGEV (**B**), p-PDCoV (**C**), and p-SADS-CoV (**D**) with different concentrations from 1.21 × 10^9^ to 1.21 × 10^2^ copies/μL (final reaction concentration: 1.21 × 10^8^ to 1.21 × 10^1^ copies/μL), respectively. The standard curves (**E**) showed excellent linear relationship (R^2^ ≥ 0.999) between logarithm of templates and their Ct values.

**Figure 2 vetsci-09-00634-f002:**
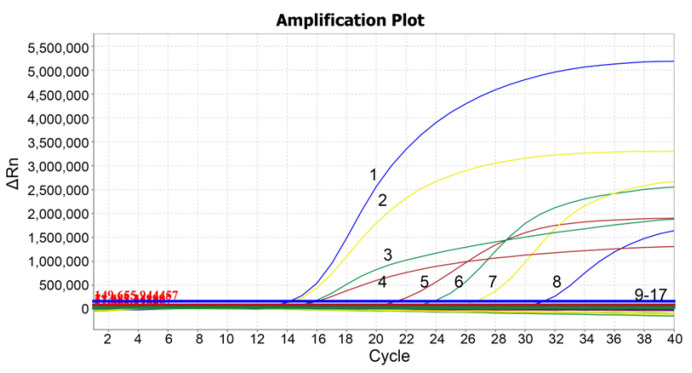
Specificity analysis. 1: p-TGEV; 2: p-PDCoV; 3: p-SADS-CoV; 4: p-PEDV; 5: PEDV; 6: SADS-CoV; 7: PDCoV; 8: TGEV; 9–16: ASFV, FMDV, CSFV, PRRSV, PRV, PoRV, PCV1, and PCV2; 17: negative control. There were specific amplification curves only for PEDV, TGEV, PDCoV, and SADS-CoV.

**Figure 3 vetsci-09-00634-f003:**
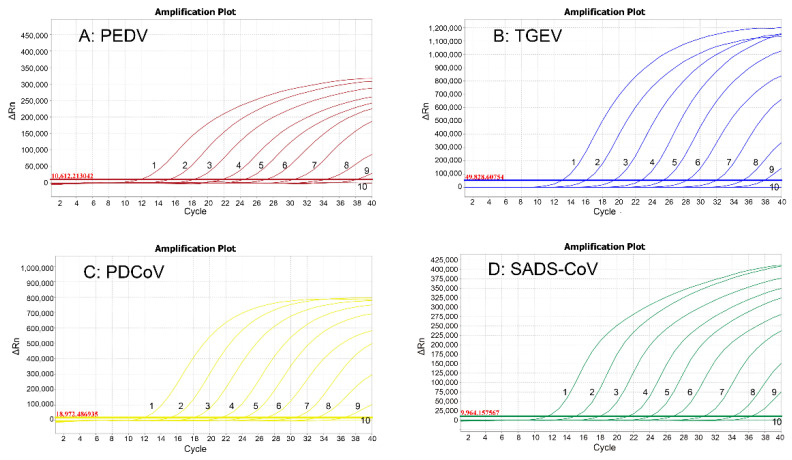
Sensitivity analysis. 1–10: 1.21 × 10^9^–1.21 × 10^0^ copies/μL (final reaction concentrations: 1.21 × 10^8^ to 1.21 × 10^−1^ copies/μL) of the standard plasmids. The limit of detection (LOD) of all PEDV, TGEV, PDCoV, and SADS-CoV was 1.21 × 10^2^ copies/μL (final reaction concentration: 1.21 × 10^1^ copies/μL).

**Table 1 vetsci-09-00634-t001:** The sequences of the primers and probes used to differentiate PEDV, TGEV, PDCoV, and SADS-CoV.

Primer/Probe	Sequence (5′→3′)	Product/bp
PEDV(N)-U	CTGGAATGAGCAAATTCGCTG	140
PEDV(N)-D	CAACCCAGAAAACACCCTCAG	
PEDV(N)-P	JOE-AGCGAATTGAACAACCTTCCAATTGGCA-BHQ1	
TGEV(M)-U	GCAATTCTTTGCGTTAGTGCAT	102
TGEV(M)-D	AGCGTACAAATTCCCTGAAAGC	
TGEV(M)-P	Texas Red-CTTCCTCTCGAAGGTGTGCCAACTGG-BHQ2	
PDCoV(M)-U	ATCGACCACATGGCTCCAA	72
PDCoV(M)-D	CAGCTCTTGCCCATGTAGCTT	
PDCoV(M)-P	FAM-CACACCAGTCGTTAAGCATGGCAAGCT-BHQ1	
SADS-CoV(N)-U	TACTGGTCCTCACGCAGATG	120
SADS-CoV(N)-D	ACGATTGCGAACACCAAGAC	
SADS-CoV(N)-P	Cy5-CAACAGCGACCCAATGCACACCCT-BHQ3	

**Table 2 vetsci-09-00634-t002:** The components of the optimal reaction system.

Reagent	Volume (µL)	Final Concentration (nM)
2× One-Step RT-PCR Buffer III (TaKaRa)	10	/
Ex Taq HS (5 U/μL) (TaKaRa)	0.4	/
PrimeScript RT Enzyme Mix II (TaKaRa)	0.4	/
PEDV(N)-U (20 pmol/μL)	0.2	200
PEDV(N)-D (20 pmol/μL)	0.2	200
PEDV(N)-P (20 pmol/μL)	0.3	300
TGEV(M)-U (20 pmol/μL)	0.3	300
TGEV(M)-D (20 pmol/μL)	0.3	300
TGEV(M)-P (20 pmol/μL)	0.3	300
PDCoV(M)-U (20 pmol/μL)	0.2	200
PDCoV(M)-D (20 pmol/μL)	0.2	200
PDCoV(M)-P (20 pmol/μL)	0.2	200
SADS-CoV(N)-U (20 pmol/μL)	0.3	300
SADS-CoV(N)-D (20 pmol/μL)	0.3	300
SADS-CoV(N)-P (20 pmol/μL)	0.3	300
Nucleic acid template	2.0	/
RNase-free distilled H_2_O	Up to 20	/

**Table 3 vetsci-09-00634-t003:** The Ct values and coefficients of variation for repeatability analysis.

Plasmid	Concentration (Copies/μL)	Intra-Assay			Inter-Assay		
		$\bar{x}$	SD	CV (%)	$\bar{x}$	SD	CV (%)
p-PEDV	1.21 × 10^9^	11.96	0.13	1.09	11.88	0.04	0.34
	1.21 × 10^7^	18.36	0.14	0.76	18.30	0.18	0.98
	1.21 × 10^5^	24.44	0.06	0.25	24.52	0.16	0.65
p-TGEV	1.21 × 10^9^	12.97	0.05	0.39	12.87	0.12	0.93
	1.21 × 10^7^	19.26	0.03	0.16	19.12	0.17	0.89
	1.21 × 10^5^	25.12	0.19	0.76	25.10	0.09	0.36
p-PDCoV	1.21 × 10^9^	11.84	0.10	0.84	11.72	0.12	1.02
	1.21 × 10^7^	18.27	0.27	1.48	18.55	0.10	0.54
	1.21 × 10^5^	24.41	0.14	0.57	24.67	0.15	0.61
p-SADS-CoV	1.21 × 10^9^	11.51	0.02	0.17	11.79	0.22	1.87
	1.21 × 10^7^	17.81	0.06	0.34	17.95	0.18	1.00
	1.21 × 10^5^	24.51	0.35	1.43	24.75	0.11	0.44

**Table 4 vetsci-09-00634-t004:** Evaluation results of the clinical samples.

Date	Number	Number of Positive Samples					
		PEDV (%)	TGEV (%)	PDCoV (%)	SADS-CoV (%)	PEDV + TGEV (%)	PEDV + PDCoV (%)
October, 2020	200	26 (13.00%)	3 (1.50%)	30 (15.00%)	0	0	15 (7.50%)
March, 2021	112	32 (28.57%)	1 (0.89%)	71 (63.39%)	0	1 (0.89%)	12 (10.71%)
October, 2021	37	6 (16.22%)	0	0	0	0	0
November, 2021	217	3 (1.38%)	0	1 (0.46%)	0	0	0
January, 2022	314	90 (28.66%)	0	64 (20.38%)	0	0	4 (1.27%)
February, 2022	394	110 (27.92%)	0	74 (18.78%)	0	0	1 (0.25%)
March, 2022	244	16 (6.56%)	0	40 (16.39%)	0	0	0
April, 2022	378	95 (25.13%)	1 (0.26%)	26 (6.88%)	0	0	2 (0.53%)
May, 2022	30	4 (13.33%)	0	0	0	0	0
June, 2022	496	58 (11.69%)	0	2 (0.40%)	0	0	0
July, 2022	287	47 (16.38%)	0	33 (11.50%)	0	0	4 (1.39%)
August, 2022	76	14 (18.42%)	2 (2.63%)	11 (14.47%)	0	0	0
September, 2022	92	23 (25.00%)	2 (2.17%)	16 (17.39%)	1 (1.09%)	0	2 (2.17%)
October, 2022	359	67 (18.66%)	6 (1.67%)	58 (16.16%)	4 (1.11%)	1 (0.28%)	6 (1.67%)
Total	3236	591 (18.26%)	15 (0.46%)	426 (13.16%)	5 (0.15%)	2 (0.06%)	46 (1.42%)

**Table 5 vetsci-09-00634-t005:** Agreements of the detection results by the developed and the reference multiplex qRT-PCR.

Method	Positive Samples			
	PEDV (%)	TGEV (%)	PDCoV (%)	SADS-CoV (%)
The developed quadruplex qRT-PCR	591/3236 (18.26%)	15/3236 (0.46%)	426/3236 (13.16%)	5/3236 (0.15%)
The reference multiplex qRT-PCR	572/3236 (17.67%)	15/3236 (0.46%)	417/3236 (12.89%)	5/3236 (0.15%)
Agreements	99.41%	100%	99.72%	100%

## Data Availability

Not applicable.

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
