# Peer review of "A Quadruplex qRT-PCR for Differential Detection of Four Porcine Enteric Coronaviruses"

_vetsci, 2022, doi:10.3390/vetsci9110634_

Round 1

Reviewer 1 Report

Review report ID- vetsci-1982125:

The manuscript submitted to Veterinary Sciences and entitled “Development of a multiplex qRT-PCR assay for differential detection of four porcine enteric coronaviruses” is an interesting analysis of enteric corona viruses and the impact that these virus have on the livestock industry. Although there are currently a number of similar multiplexing papers published recently using clinical samples in China, the virus combination used in the current study is different from previous studies.  Perhaps the most value of this paper lies in the evaluation of the 2422 clinical fecal samples from Guangxi province in China. This paper is therefore an important contribution and will be of general interest to researchers in the field. Listed below are some comments and suggestions for its improvement.

Abstract

Line 16: Would it not be better to say of 121 copies/µl instead of 1.21 × 102 copies/μL and (final reaction concentration of 12.1 copies/μL instead of 1.21 × 101 copies/μL) for each virus.

Language editing:

Line 16: “with the lest detection limit” change to “with the lowest detection limit”

Line 30: Should be “genomes”

Line 32: Should be “which include transmissible gastroenteritis”

Line 37-38: Should be “to neonatal piglets, characterized by loss of weight, diarrhea, vomiting, and dehydration”

Line 40: Should be: “strain outbreak in China”

Line 49: Should be: “in other countries until now”

Line 225: Should be: “The assay could”

Line 283: Should be: “which raises more”

This is just a few of the edits, please check the paper.

Author Response

Reviewer 1:

Comments and Suggestions for Authors

Review report ID- vetsci-1982125:

The manuscript submitted to Veterinary Sciences and entitled “Development of a multiplex qRT-PCR assay for differential detection of four porcine enteric coronaviruses” is an interesting analysis of enteric corona viruses and the impact that these viruses have on the livestock industry. Although there are currently a number of similar multiplexing papers published recently using clinical samples in China, the virus combination used in the current study is different from previous studies.  Perhaps the most value of this paper lies in the evaluation of the 2422 clinical fecal samples from Guangxi province in China. This paper is therefore an important contribution and will be of general interest to researchers in the field. Listed below are some comments and suggestions for its improvement.

Abstract

  1. Line 16: Would it not be better to say of 121 copies/µl instead of 1.21 × 102 copies/μL and (final reaction concentration of 12.1 copies/μL instead of 1.21 × 101 copies/μL) for each virus.

Response: We agree the reviewer’s suggestion. Change “1.21 × 102 copies/μL” to “121 copies/µL”, and Change “1.21 × 101 copies/μL” to “12.1 copies/μL”. Please see lines 25-26 in the revised manuscript.

Language editing:

  1. Line 16: “with the lest detection limit” change to “with the lowest detection limit”

Response: Change “with the lest detection limit” to “with the limit of detection (LOD)”. Please see line 25 in the revised manuscript.

  1. Line 30: Should be “genomes”

Response: Change “genome” to “genomes”. Please see line 39 in the revised manuscript.

  1. Line 32: Should be “which include transmissible gastroenteritis”

Response: Change “which including” to “which include”. Please see line 41 in the revised manuscript.

  1. Line 37-38: Should be “to neonatal piglets, characterized by loss of weight, diarrhea, vomiting, and dehydration”

Response: Change “which characterizing with loss of weight, diarrhea, vomiting, and dehydration” to “characterized by loss of weight, diarrhea, vomiting, and dehydration”. Please see lines 46-47 in the revised manuscript.

  1. Line 40: Should be: “strain outbreak in China”

Response: “outbroke” is correct. Please see line 49 in the revised manuscript.

  1. Line 49: Should be: “in other countries until now”

Response: Change “in other country until now” to “in other countries until now”. Please see line 59 in the revised manuscript.

  1. Line 225: Should be: “The assay could”

Response: Change “The assay couuld” to “The assay could”. Please see line 242 in the revised manuscript.

  1. Line 283: Should be: “which raises more”

Response: Change “where raises more” to “which raises more”. Please see line 299 in the revised manuscript.

  1. This is just a few of the edits, please check the paper.

Response: We also checked the manuscript carefully and revised some mistakes. Please see the revised manuscript.

Reviewer 2 Report

This manuscript from Zhou et al details the development and performance of a quadruplex RT-qPCR assay for identifying and quantifying four coronavirus species (PEDV, TGEV, PDCoV and SADS-CoV), which are leading causes of porcine gastrointestinal disease, in porcine fecal samples.  Their protocol-optimized primer and probe sets demonstrated appropriate amplification efficiency, sensitivity and repeatability, and no non-specific amplification of other porcine viruses.  This assay will be useful for ongoing surveillance of porcine coronaviruses.  My only comment to be addressed prior to acceptance for publication is to report the results of repeating sections 3.4 and 3.5, as demonstrated repeatability closer the limits of detection (1.21 * 10^2 copies/ uL) as well as with ensuring the lack of non-specific amplification were not included in section 3.6.

Author Response

Reviewer 2:

Comments and Suggestions for Authors

This manuscript from Zhou et al details the development and performance of a quadruplex RT-qPCR assay for identifying and quantifying four coronavirus species (PEDV, TGEV, PDCoV and SADS-CoV), which are leading causes of porcine gastrointestinal disease, in porcine fecal samples.  Their protocol-optimized primer and probe sets demonstrated appropriate amplification efficiency, sensitivity and repeatability, and no non-specific amplification of other porcine viruses.  This assay will be useful for ongoing surveillance of porcine coronaviruses.  My only comment to be addressed prior to acceptance for publication is to report the results of repeating sections 3.4 and 3.5, as demonstrated repeatability closer the limits of detection (1.21 * 10^2 copies/uL) as well as with ensuring the lack of non-specific amplification were not included in section 3.6.

Response: In this study, the specificity analysis was showed in section 3.4, the sensitivity analysis was showed in section 3.5, and the repeatability analysis was showed in section 3.6. The results showed that the developed quadruplex qRT-PCR illustrated extreme specificity, high sensitivity, and excellent repeatability. Therefore, we add these characteristics of the developed assay in the revised manuscript. Please see lines 194-195, 207, and 218-219 in the revised manuscript.

Reviewer 3 Report

1.     Line 15, delete “lest”

2.     Line 50, change “Chinese Hong Kong” to “Hong Kong, China”

3.     Line 67-70, what's the advantage of this multiplex qRT-PCR compared to the existing methods? Please include the explanation in the Introduction part. 

4.     Line 225, change “could” to “could”

5.     Line 226, change “had” to “and had”

Author Response

Reviewer 3:

Comments and Suggestions for Authors

  1. Line 15, delete “lest”

Response: Change “with the lest detection limit” to “with the limit of detection (LOD)”. Please see line 25 in the revised manuscript.

  1. Line 50, change “Chinese Hong Kong” to “Hong Kong, China”

Response: change “Chinese Hong Kong” to “Hong Kong, China”. Please see line 60 in the revised manuscript.

  1. Line 67-70, what's the advantage of this multiplex qRT-PCR compared to the existing methods? Please include the explanation in the Introduction part.

Response: The advantages of the multiplex qRT-PCR were described in the Introduction part. Please see lines 77-79 in the revised manuscript.

  1. Line 225, change “could” to “could”

Response: Change “couuld” to “could”. Please see line 242 in the revised manuscript.

  1. Line 226, change “had” to “and had”

Response: Change “had” to “and had”. Please see line 242 in the revised manuscript.

Reviewer 4 Report

Dear Authors.

I find the date you present do not support the statements you made. The sample collection does not include variety and the lack of SADS-CoV invalidate some of your claims, then credibility on the paper. I would suggest to reconsider include more samples on the sample set. From other geographical areas, and with SADS-CoV positive samples. If not possible, they will have to be spiked (and declared) but it needs to be tested. The specificity and sensibility testing need to be improved. The current document is not right and the conclusions overstep your findings. That’s not rigurous and that can affect your credibility. I would recommend to improve the study in order to prevent any credibility damage.

The sample collection you used for the testing had a remarkable number of samples, but there not information about the number of farms involved and it certainly come for a restricted geographical area. Either if the strains detected are close related in the phylogeny (for the primer target genes) or not, even if they are well conserve regions, that add some doubts on the performance data when other strains would be involved. This type of studies generally include samples from different distant regions and/or samples of strains that are known to be distantly related (for the target genes if possible). Your study lacks of these aspects. It may perform as you calculated in that region with that samples, but that figure may be different in other areas, or when other strains of these pathogens become present in the region. Out of the different pathogens you consider, 2 of them are mostly present in East Asia, therefore your samples may provide a good scope of the genetic variability, perhaps not. In the case of PEDV (or TGVE) they are spread worldwide and variability is continuously increasing. Therefore, I think you have to readjust the statements all though the document as the figures related to the performance of the test could be notably biased by these facts; perhaps you can add prove of the contrary or explain why is not like that. Otherwise, the statements to acknowledge that limitation and clarify that these figures just describe the performance of the multiplex in that region in this time. It may not be so probably for PEDV where you based the design on an European strain (but quite old though), but the TGEV sequence came from a regional strain if I am not wrong.

The fact that no clinical samples contained SADS-COv invalidate some of the most important statements in the Discussion and Conclusion. You have prove it works for 3 pathogens with strains of that regions, you have not proved it works for the 4 or everywhere.

LINE 32: The sentence reads as each of the 6 coronavirus produce respiratory and gastrointestinal disease, in my knowledge they do not. Please, could you rephrase.

LINE 72 to 78: I am totally lost with this vaccine section. Why FMVD, CSF, PRRSV, PRV and PCV2 vaccines are mentioned. It is not understood till you read line 123, so it can be added to the paragraph 2.7. Abbreviations are not described. The last sentence about the plasmid is not connected to previous text, so out of context. I would suggest to restructure the text to make that sentence fit once some previous explanations were available.

LINE 80: I found more valuable to know if the 2422 samples come from 10, 100 or 1000 farms as samples from the same unit will be closely related.

LINE 123: Why the material for the specificity analysis was obtained from the vaccines instead of form the pathogens DNA/RNA. Did you verify there were not traces of vaccine adjuvant and excipients inhibiting the PCR reaction? Why the specificity panel did not included other swine virus of the coronaviridae family that are not the multiplex panel? Why HEV - beta coronavirus or Torovirus were not included in the specificity testing instead of the other habitual but unrelated virus?

LINE 177: Did you carried out a control test to verify if the material containing FMD, CSF… from the vaccines still allowed the produce a positive response in the PCR or some molecule (adjuvant, excipient) from the vaccine could be inhibiting it?

LINE 204 (point 3.7) If all the clinical samples were negative for SADs-CoV. How could you later conclude the multiplex performs well for this type of pathogen. I guess it reacted to your positive control, but has it been tested with other strains? Has it detected SADS-Cov in a clinical sample?

LINE 234: You say the multiplex is useful because it agrees 99% with the other test but it is unfair to use just one figure for all instead of a figure for each pathogen. The truth is that that SADS-CoV was not present invalidates the statement. You can only state it works for 3 of them and you really unknown which is the level of agreement for SADS-CoV.

LINE 289-290. This study do not demonstrate the multiplex can detect the four viruses on clinical samples for the reasons mentioned before.

Author Response

Reviewer 4:

Comments and Suggestions for Authors

Dear Authors.

  1. I find the date you present do not support the statements you made. The sample collection does not include variety and the lack of SADS-CoV invalidate some of your claims, then credibility on the paper. I would suggest to reconsider include more samples on the sample set. From other geographical areas, and with SADS-CoV positive samples. If not possible, they will have to be spiked (and declared) but it needs to be tested. The specificity and sensibility testing need to be improved. The current document is not right and the conclusions overstep your findings. That’s not rigurous and that can affect your credibility. I would recommend to improve the study in order to prevent any credibility damage.

Response: In this study, a quadruplex qRT-PCR was developed for detection of PEDV, TGEV, PDCoV, and SADS-CoV, and a total of 2422 clinical samples was used to validate the application of this assay. However, no positive sample of SADS-CoV was found in these samples. SADS-CoV was ever discovered in Guangxi province, China, but the positive rate was very low (please see reference 17). In our study, the clinical samples were collected from 179 different pig farms in Guangxi province between October 2020 and June 2022. According to the results in this study, we need continuously collect clinical samples from more pig farms to monitor SADS-CoV in case of its emergence and re-emergence in Guangxi province. In the revised manuscript, the number of pig farms was added. Please see line 95 in the revised manuscript.

The 2422 clinical samples were tested by the developed quadruplex qRT-PCR in this study, and also tested by the reference method reported previously by Huang et al (please see reference 37). No positive sample of SADS-CoV was discovered by these two methods, showing that their results were consistent. In the previous report, a total of 354 clinical samples were tested by the established qRT-PCR, and 5 samples were positive for SADS-CoV. Therefore, we believe that it is acceptable to use the reference method as standard method, and the results of the developed qRT-PCR were accurate and credible.

  1. The sample collection you used for the testing had a remarkable number of samples, but there not information about the number of farms involved and it certainly come for a restricted geographical area. Either if the strains detected are close related in the phylogeny (for the primer target genes) or not, even if they are well conserve regions, that add some doubts on the performance data when other strains would be involved. This type of studies generally include samples from different distant regions and/or samples of strains that are known to be distantly related (for the target genes if possible). Your study lacks of these aspects. It may perform as you calculated in that region with that samples, but that figure may be different in other areas, or when other strains of these pathogens become present in the region. Out of the different pathogens you consider, 2 of them are mostly present in East Asia, therefore your samples may provide a good scope of the genetic variability, perhaps not. In the case of PEDV (or TGVE) they are spread worldwide and variability is continuously increasing. Therefore, I think you have to readjust the statements all though the document as the figures related to the performance of the test could be notably biased by these facts; perhaps you can add prove of the contrary or explain why is not like that. Otherwise, the statements to acknowledge that limitation and clarify that these figures just describe the performance of the multiplex in that region in this time. It may not be so probably for PEDV where you based the design on an European strain (but quite old though), but the TGEV sequence came from a regional strain if I am not wrong.

Response: In our study, the clinical samples were collected from 179 different pig farms in Guangxi province between October 2020 and June 2022. The number of pig farms was added in the revised manuscript. Please see line 95 in the revised manuscript. According to our study and other scientists’s study, the circulating strains of PEDV, TGEV, PDCoV, and SADS-CoV had similar genetic characterization with those from other provinces in China, and the clinical samples from different regions of Guangxi province were suitable to be used for validating the application of the developed assay.

After comparing the viral genomic sequences downloaded from NCBI GenBank, the conserved genomic regions of PEDV, TGEV, PDCoV, and SADS-CoV were selected as the target regions for designing the specific primers and corresponding probes. The viral strains were from different countries, not limiting to the European strains or the regional strains. Therefore, the designed primers and probes were universal for detection of the targeted viruses.

  1. The fact that no clinical samples contained SADS-COv invalidate some of the most important statements in the Discussion and Conclusion. You have prove it works for 3 pathogens with strains of that regions, you have not proved it works for the 4 or everywhere.

Response: The 2422 clinical samples were tested by the developed qRT-PCR in this study, and also tested by the reference method reported previously by Huang et al (please see reference 37). No positive sample of SADS-CoV was discovered by these two methods in our present study, showing that their results were consistent. In the previous report, a total of 354 clinical samples were tested by the established multiplex qRT-PCR, and 5 samples were positive for SADS-CoV. Therefore, we believe that it is acceptable to use the reference method as standard method, and the results of the developed quadruplex qRT-PCR were accurate and credible. Of course, we should collect more samples to test these viruses in the future, especially SADS-CoV, in order to further validate the application of this assay.

  1. LINE 32: The sentence reads as each of the 6 coronavirus produce respiratory and gastrointestinal disease, in my knowledge they do not. Please, could you rephrase.

Response: Change “cause respiratory illness and gastroenteritis in pigs” to “cause respiratory illness or gastroenteritis in pigs”. Please see line 41 in the revised manuscript.

  1. LINE 72 to 78: I am totally lost with this vaccine section. Why FMVD, CSF, PRRSV, PRV and PCV2 vaccines are mentioned. It is not understood till you read line 123, so it can be added to the paragraph 2.7. Abbreviations are not described. The last sentence about the plasmid is not connected to previous text, so out of context. I would suggest to restructure the text to make that sentence fit once some previous explanations were available.

Response: We agree the reviewer’s suggestions. The first paragraph of section 2.1 has been rewritten. The full names of the viruses and their abbreviations were described when they first appeared in the manuscript. Please see line 85-94 in the revised manuscript.

  1. LINE 80: I found more valuable to know if the 2422 samples come from 10, 100 or 1000 farms as samples from the same unit will be closely related.

Response: The clinical samples were collected from 179 different pig farms in Guangxi province between October 2020 and June 2022. The number of pig farms was added in the revised manuscript. Please see line 95 in the revised manuscript.

  1. LINE 123: Why the material for the specificity analysis was obtained from the vaccines instead of form the pathogens DNA/RNA. Did you verify there were not traces of vaccine adjuvant and excipients inhibiting the PCR reaction? Why the specificity panel did not included other swine virus of the coronaviridae family that are not the multiplex panel? Why HEV - beta coronavirus or Torovirus were not included in the specificity testing instead of the other habitual but unrelated virus?

Response: The live or inactivated vaccine strain of PEDV, TGEV, PoRV, FMDV, CSFV, PRRSV, PRV, and PCV2 were easy to obtained, and their wild-type strains were hard to obtained in China. For the sake of biosafety, we prefer to use vaccine strains for test operations. After extraction of the viral DNAs or RNAs from the vaccines, these viruses were tested by the corresponding qPCR or qRT-PCR, respectively, and confirmed to show typical amplification curves. Therefore, they were used for the specificity analysis of the developed assay.

In this study, ASFV, PoRV, FMDV, CSFV, PRRSV, PRV, PCV1, and PCV2 were used for the specificity analysis of the developed assay. These viruses are the important viruses that are circulating in the field in China, and therefore were selected as control viruses. HEV and Torovirus were not included in the specificity test in this study.

  1. LINE 177: Did you carried out a control test to verify if the material containing FMD, CSF… from the vaccines still allowed the produce a positive response in the PCR or some molecule (adjuvant, excipient) from the vaccine could be inhibiting it?

Response: After extraction of the viral DNAs or RNAs from the vaccines, these viruses were tested by the corresponding qPCR or qRT-PCR, respectively, and confirmed to show typical amplification curves. Therefore, they were used for the specificity analysis of the developed assay.

  1. LINE 204 (point 3.7) If all the clinical samples were negative for SADs-CoV. How could you later conclude the multiplex performs well for this type of pathogen. I guess it reacted to your positive control, but has it been tested with other strains? Has it detected SADS-Cov in a clinical sample?

Response: Really, since all the clinical samples were negative for SADS-CoV, we used the synthetic positive plasmids of SADS-CoV to evaluate the developed quadruplex qRT-PCR. Furthermore, the results tested by the developed method were confirmed by the reference method reported by Huang et al. (please see reference 37). Certainly, we will continuously monitor this pathogen in Guangxi province using the developed method and other methods in case of its re-emergence.

  1. LINE 234: You say the multiplex is useful because it agrees 99% with the other test but it is unfair to use just one figure for all instead of a figure for each pathogen. The truth is that that SADS-CoV was not present invalidates the statement. You can only state it works for 3 of them and you really unknown which is the level of agreement for SADS-CoV.

Response: The 2422 clinical samples were tested by the developed quadruplex qRT-PCR in this study, and also tested by the reference method reported previously by Huang et al. (please see reference 37). The tested results of PEDV, TGEV, PDCoV, and SADS-CoV were consistent by these two methods, and the compliance rates were more than 99%. Especially, no positive sample of SADS-CoV was discovered by these two methods in our present study, showing that their results were consistent. In the previous report by Huang et al. (please see reference 37), a total of 354 clinical samples were tested by the established quadruplex qRT-PCR, and 5 samples were positive for SADS-CoV. Therefore, we believe that it is acceptable to use the reference method as standard method, and the results of the developed quadruplex qRT-PCR in this study were accurate and credible. Of course, we should collect more samples from different pig farms to test these viruses in the future, especially SADS-CoV, in order to further validate the application of this assay.

  1. LINE 289-290. This study do not demonstrate the multiplex can detect the four viruses on clinical samples for the reasons mentioned before.

Response: The 2422 clinical samples were tested by the developed quadruplex qRT-PCR in this study, and also tested by the reference method reported previously by Huang et al. (please see reference 37). The tested results of PEDV, TGEV, PDCoV, and SADS-CoV were consistent by these two methods, and the compliance rates were more than 99%, It is worth to mentioned that no positive sample of SADS-CoV was discovered by these two methods in our present study, showing that their results were consistent. In the previous report by Huang et al. (please see reference 37), a total of 354 clinical samples were tested by the established qRT-PCR, and 5 samples were positive for SADS-CoV. Therefore, we believe that it is acceptable to use the reference method as standard method, and the results of the developed It is worth mentioned qRT-PCR were accurate and credible. Of course, we should continuously collect more samples from different pig farms to test these viruses in the future, especially SADS-CoV, in order to further validate the application of this assay.

Round 2

Reviewer 4 Report

Dear authors. 

Your explanation and the new data you added represent a major change in this article. 

Regarding rebuttal number 2, I complained about the lack of non-regional samples in the data set, so you were not able to provide proof the pcr was suitable to work with other strains. You answered you tested it in-silico but the actual PCR is the only way to prove it actually works. Please, could you explain why we should accept those in silico results? Which reason do you argue to make those results acceptable instead of modifying your document and stating the validation was only done on local samples and you cannot ensure performance in other strains? Indeed, you have no data about phylogeny in your strains, do you?

Table 4 – “December 22” may be a type. I guess it is “September 22”. Please, could you specify if the samples you added belong to Huang et all (ref 37) collection?

I personally think the fact that HEV and Torovirus -but other unrelated common virus instead- were not tested in the specificity test is something that should be reported. Though, it is up to you to do it or not. 

Best regards

Author Response

9 November, 2022

Revision notes

  1. Regarding rebuttal number 2, I complained about the lack of non-regional samples in the data set, so you were not able to provide proof the pcr was suitable to work with other strains. You answered you tested it in-silico but the actual PCR is the only way to prove it actually works. Please, could you explain why we should accept those in silico results? Which reason do you argue to make those results acceptable instead of modifying your document and stating the validation was only done on local samples and you cannot ensure performance in other strains? Indeed, you have no data about phylogeny in your strains, do you?

Response: We downloaded the sequences of different strains of PEDV, TGEV, PDCoV, and SADS-CoV from NCBI GenBank. These sequences came from China, and also from other countries. Of the downloaded sequences from China, some sequences came from our laboratory. Then, we analyzed the results of sequence alignment of these strains, and selected the conserved regions as the targeted regions for designing primers and probes. The primers and probes were analyzed using GenBank blast, and confirmed to be specific for PEDV, TGEV, PDCoV, and SADS-CoV. Furthermore, we used different viruses, including ASFV, FMDV, CSFV, PRRSV, PRV, PoRV, PCV1, and PCV2 to evaluate the specificity of the primers and probes. In our study, the clinical samples were collected from Guangxi province, China to confirm the application of the developed quadruplex qRT-PCR. According to the above results, the specific primers and probes basing on the conserved regions were suitable for different strains of PEDV, TGEV, PDCoV, and SADS-CoV from different countries.

Since our instutution is a regional (provincal) organization, we collected the clinical samples from different pig farms within Guangxi province. However, it is very common and frequent for the trade of pigs and pig products among different provinces in China. The results of molecular epidemiological study in our laboratory and in other laboratories have indicated that the sequences of these viruses from different provinces in China were similar, and there existed conserved regions among different strains, even if there existed minor variation and genetic diversity of different strains. Beside the silico results, the developed quadrepluex qRT-PCR was used to tested the clinical samples in order to validate the application of this assay. Therefore, we believe the developed assay is suitable to work with other strains beside our strains. Of course, there are so many different strains in different countries, and even in different provinces in China, it is impossible for us to obtain all of them to evaluate the developed assay, and this is unnecessory. In the Discussion, we discused this issue in the revised manuscript. Please see Lines 317-321 in the revised manuscript.

We have been doing the epidemiological study on PEDV, TGEV, PDCoV, and SADS-CoV in Guangxi province, China for sereral years. We have obtained many genomic sequences of these viruses, and have data about phylogeny in our strains. Now, we are preparing the manuscript on the genomic characterization of PEDV, and PDCoV in Guangxi province, China. In addition, our several papers have been published as follows:

(1) Huang H, Yin Y, Wang W, Cao L, Sun W, Shi K, Lu H, Jin N. Emergence of Thailand-like strains of porcine deltacoronavirus in Guangxi Province, China. Vet Med Sci. 2020, 6(4): 854-859. doi: 10.1002/vms3.283.

(2) Yan J, Shi K, Li Z, Yin Y, Lu W, Qu S. Genetic diversity of porcine deltacoronavirus in Guangxi Province from 2017 to 2019. Chinese Veterinary Science, 2020, 50(09): 1147-1158. (in chinese)

(3) Wang R, Shi K, Yan J, Xie S, Li Z, Yin Y, Lu W, Qu S. Genetic diversity of porcine epidemic diarrhea virus in Guangxi Province from 2017 to 2019. Chinese Veterinary Science, 2020, 50(02): 189-198. (in chinese)

The phyolyogenetic trees of PEDV, TGEV, and PDCoV are showed as follows.

Figure 1. Phylogenetic analyses of PDCoV based on the complete genome (a) and S gene (b) using the neighbour-joining algorithm and a heuristic search with 1,000 bootstrap replications implemented in MEGA7

Figure 2. Phylogenetic tree based on nucleotide sequences of S gene (Left) and M gene (Right)

  1. Table 4 – “December 22” may be a type. I guess it is “September 22”. Please, could you specify if the samples you added belong to Huang et all (ref 37) collection?

Response: Really, “December, 2022” should be “September, 2022”. According the information of the pig farms from the authors of the reference 17, we collected the clinical samples from those pig farms and other pig farms nearby, and 5 samples were positive for SADS-CoV.

  1. I personally think the fact that HEV and Torovirus -but other unrelated common virus instead- were not tested in the specificity test is something that should be reported. Though, it is up to you to do it or not. 

Response: In this study, ASFV, FMDV, CSFV, PRRSV, PRV, PoRV, PCV1, and PCV2 were used to evaluate the specificity of the developed quadruplex qRT-PCR. These viruses are the common pathogens circulating in the field in China, so they were used as the control viruses for specificity analysis of the developed assay. We did not use HEV and Torovirus to evaluate the specificity of the developed assay. We discussed this issue in the revised manuscript in the Discussion part. Please see Lines 311-317 in the revised manuscript.

Best regards

Kaichuang Shi
